# Self-sorting in macroscopic supramolecular self-assembly via additive effects of capillary and magnetic forces

Minghui Tan [1], Pan Tian [1], Qian Zhang [2], Guiqiang Zhu [2], Yuchen Liu [3], Mengjiao Cheng [1,2,3] ✉ & Feng Shi [1,2,3] ✉

Supramolecular self-assembly of μm-to-mm sized components is essential to construct complex supramolecular systems. However, the selective assembly to form designated structures at this length scale is challenging because the short-ranged molecular recognition could hardly direct the assembly of macroscopic components. Here we demonstrate a self-sorting mechanism to automatically identify the surface chemistry of μm-to-mm components (A: polycations; B: polyanions) based on the A-B attraction and the A-A repulsion, which is realized by the additivity and the competence between long-ranged magnetic/capillary forces, respectively. Mechanistic studies of the correlation between the magnetic/capillary forces and the interactive distance have revealed the energy landscape of each assembly pattern to support the self-sorting results. By applying this mechanism, the assembly yield of ABA trimers has been increased from 30%~40% under conventional conditions to 100% with self-sorting. Moreover, we have demonstrated rapid and spontaneous self-assembly of advanced chain-like structures with alternate surface chemistry.

Supramolecular assembly across all length scales is meaningful for the construction of complex functional systems[1-7], which has stimulated diverse concepts/methods such as programmable assembly[8,9], pathway-selective assembly[10,11], self-sorting[12,13]. As the assembly scales up with many pathways and meta-stable structures, the challenge to selectively obtain designated structures is increasing together with the problem of low assembly precision[14]. Generally, two strategies have been proposed to confine the assembly results: (1) subtle designs of molecules or nanoparticles to exclude meta-stable assemblies[15,16], which however requires tedious and difficult fabrication of microscale building blocks; (2) directly using μm-to-mm sized components to shorten the length gap from building blocks to final structures and to reduce the assembly possibilities, which is termed as macroscopic supramolecular assembly (MSA)[2,3,17-24]. Owing to the feasibility of fabrication at this length scale, it is facile to design MSA building blocks

and to adjust assembly conditions[25,26]. One typical example is inducing the design of wettability conflict at interfaces of immiscible liquids (e.g., air/water): the side surfaces of building blocks are modified with alternate wettability of hydrophilic-hydrophobic to result in oppositely shaped menisci (concave or convex)[25,27-29]; as building blocks approach, the surfaces with similarly shaped menisci merged to realize assembly while those with oppositely shaped menisci do not assemble due to capillary repulsion. The assembly precision of the above capillary-driven MSA is satisfactory with assembled surfaces well-matched; however, these MSA methods only demonstrated the selectivity in the surface wettability; a higher-levelled selective assembly, i.e., 'the self-sorting of surface chemistry', is impossible. When building blocks with the same wettability but different surface chemistry approach (e.g., A modified with host molecules and B with guest molecules), three assemblies of AA, AB and BB will all form due

[1]State Key Laboratory of Chemical Resource Engineering, Beijing University of Chemical Technology, Beijing 100029, China. [2]Beijing Laboratory of Biomedical Materials, Beijing University of Chemical Technology, Beijing 100029, China. [3]Beijing Advanced Innovation Centre for Soft Matter Science and Engineering, Beijing University of Chemical Technology, Beijing 100029, China. ✉e-mail: chengmj@mail.buct.edu.cn; shi@mail.buct.edu.cn

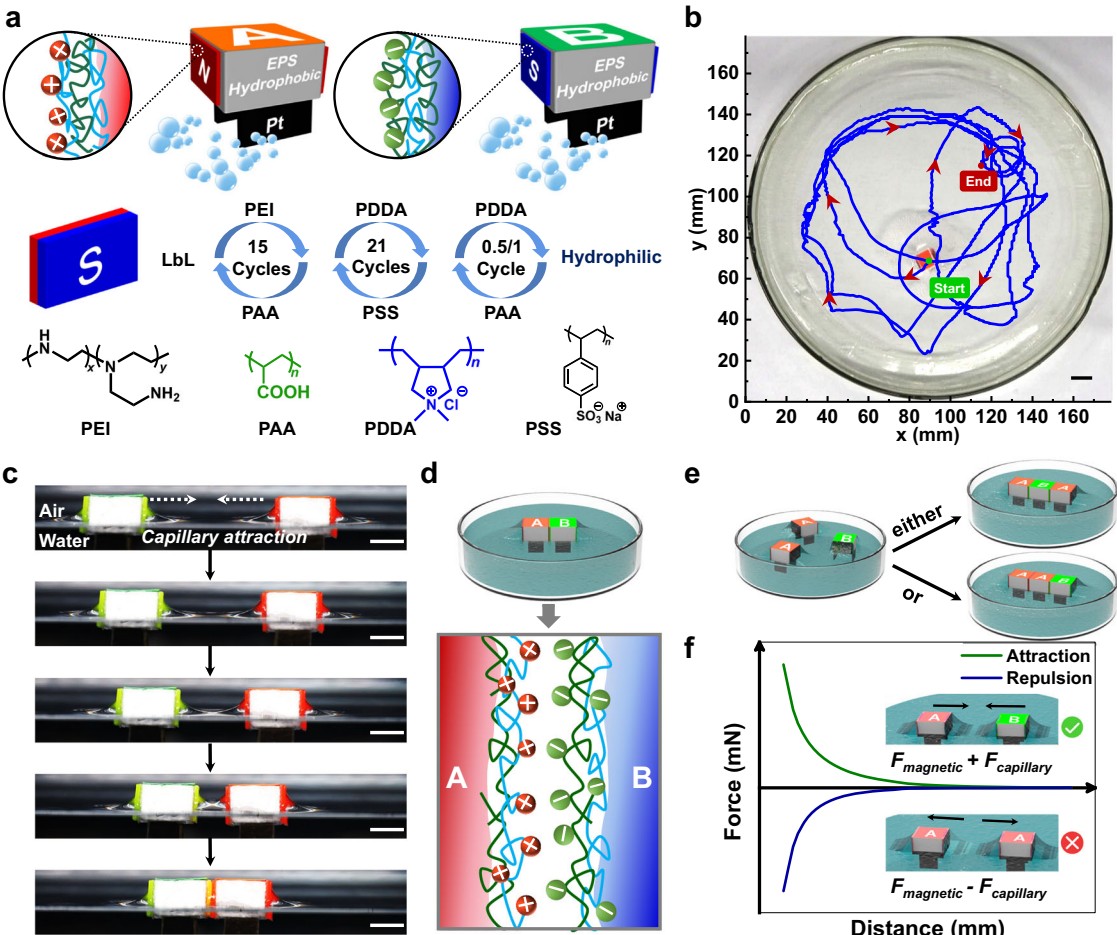

**Fig. 1 | Design of MSA building blocks and the challenge of self-sorting in macroscopic self-assembly. a** Schematic illustration of expanded polystyrene (EPS) building blocks with self-propulsive systems (catalyst of platinum, Pt), anisotropic wettability and magnetic plates. **b** Motion trajectory of one building block on the surface of $H_2O_2$ solution. Scale bar: 1 cm. **c** Snapshot photographs of A-B assembly driven by the capillary attraction at an air/water interface. Scale bars:

5 mm. **d** Interfacial electrostatic interaction between oppositely charged polyelectrolytes on A-B surfaces after capillary-driven assembly. **e** The problem of 'isomers' present in the assembly of trimers, and **f** a possible solution of directed assembly via resultant forces from the coupled magnetic ($F_{magnetic}$) and capillary ($F_{capillary}$) interactions. Source data are provided as a Source Data file.

to the capillary attraction of similar menisci. It is almost impossible to specifically obtain AB while excluding non-specific AA or BB simply by capillary-driven assembly. Although molecular interactions (e.g., host/guest recognition) are chemically specific[30,31], these molecular-level interactions are too short-ranged to align μm-to-mm objects and the assembly precision remains poor with interfacial mis-matching[2,8,17,18].

Like strategies of integrating multiple molecular interactions in molecular self-assembly to confine the assembly[4,11,32], coupling multiple long-ranged forces with a comparable strength may realize selective MSA, which has not been demonstrated yet. Magnetic forces are as long-ranged as capillary interactions with inherently N-S attraction and N-N or S-S repulsion. Although magnetic-force-driven assembly has been reported[33–35], the global effect of magnetic field remains to be overcome and a local alignment mechanism is lacking: most individual components are exerted with magnetic forces to inevitably form non-specific clusters. Therefore, neither magnetic-only nor capillary-only methods could realize the selective assembly. Herein, we have demonstrated a self-sorting strategy of MSA by coupling the effects of global magnetic sorting and local capillary aligning, and achieved chemically-specific assembly of AB structures with an assembly yield of 100%. Specifically, we have designed millimeter self-propulsive building blocks of cuboid expanded polystyrene (EPS) foam with wettability conflicts and magnets on side surfaces. The surface chemistry of hydrophilic side surfaces is identified by the orientation of N or S poles:

A and B modified with polycations and polyanions, have N poles and S poles facing outwards, respectively. As a result, A-A or B-B repelled each other upon approaching due to the competition of the stronger N-N or S-S repulsion than the hydrophilic attraction; meanwhile, A-B attract each other owing to the additivity of magnetic/capillary attraction. The measurements, calculations and simulations of magnetic/capillary forces depending on the approaching distance have revealed the energy landscape of each assembly pattern to support the above self-sorting results. By applying this mechanism, we increase the assembly yield of ABA trimers from around 30%-40% to 100% and obtained chemically-specific advanced structures of tetramer, hexamer and octamer with 'ABAB...' sequences. The self-sorting strategy has addressed the long-existing problem of selective MSA and thus filled the gap of ordered assembly at all length scales, which may facilitate precise self-assembly for complex fabrication with a massive and parallel advantage.

## Results

### Design and fabrication of assembling components

As schematically illustrated in Fig. 1a, we designed millimeter-sized cuboid EPS building blocks (Supplementary Fig. 1) to realize the processes of self-propulsive motions, self-alignment and interfacial binding similar to molecular self-assembly. The first design of self-propulsion is to mimic the diffusion and collision processes of

molecules with a bubble-driven method[27,36,37] because molecular thermal motions are not strong enough to propel µm-to-mm components. Hence, we inserted two glass slides deposited with rough platinum aggregates (Supplementary Fig. 1a, b) at the bottom surface of EPS; upon placed onto $H_2O_2$ solutions, the platinum catalyzed the decomposition of $H_2O_2$ for vigorous release of oxygen bubbles, which exerted asymmetrical forces on opposite sides and resulted in self-propulsion of EPS with a random motion trajectory (Supplementary Movie 1, Fig. 1b). The second design of capillary self-alignment was achieved by the wettability conflicts (Supplementary Fig. 2): two opposite side surfaces of EPS were modified with a fluorinated coating to exhibit hydrophobicity[38]; the other two side surfaces were adhered with hydrophilic magnetic plates that were pre-modified with polyelectrolyte multilayers[18]. The capillary-driven assembly of two hydrophilic surfaces were observed in Fig. 1c: as the building blocks approach into proximity, the menisci gradually contact and merge to eliminate excessive surface areas, which is energy-favorable to minimize the interfacial free energy. Any displacements between the two hydrophilic surfaces could be avoided because the adjacent hydrophobic surfaces with an oppositely-shaped meniscus generated a capillary repulsion to hinder displacements. Even though the capillary attraction between the two hydrophobic surfaces might also be the driving force of assembly, the curvature forces produced by the menisci of hydrophobic surfaces were weak because the light-weighted building blocks (density: ~0.47 g/cm³) deform the air/water interface little, leading to a weaker capillary force than that by hydrophilic surfaces[25]. The third design of interfacial binding after the contact of hydrophilic surfaces is to use multivalent electrostatic interactions between polycations and polyanions[39] modified on the assembled surfaces (Fig. 1d). To obtain free-standing stable assemblies immediately after the capillary-driven assembly, we applied the previously developed concept of 'flexible spacing coating' beneath the charged groups to facilitate the multivalent binding upon rapid contacting[29,39–41]. This coating was made from the composite polyelectrolyte multilayers (Fig. 1a) via a layer-by-layer (LbL) assembled technique (Supplementary Fig. 3). Without this coating, simply modifying interactive groups could hardly form stable assemblies in such short and dynamic contacting process.

## Self-sorting of AB and AA assemblies

Although the selective assembly of hydrophilic surfaces was realized, the mixture of AB, AA and BB could not be identified from capillary forces, leading to the problem of 'isomers' in the assembly of trimers (AAB and ABA) (Fig. 1e). The underlying challenge is to identify the surface chemistry of positive (A) or negative (B) polyelectrolytes on the hydrophilic surfaces. However, the selectivity of the electrostatic attraction between A-B and the electrostatic repulsion between A-A, can not distinguish these assemblies of AB or AA at macroscopic scales because the molecular-leveled electrostatic interactions are much weaker than the long-ranged capillary forces. One possible solution is to build a selective mechanism at the µm-to-mm scale by coupling the capillary interactions with another long-ranged force, e.g., magnetic force (Fig. 1f). As shown in Fig. 2a: the lateral forces between A-B include the N-S magnetic attraction and the capillary attraction, which generate additive effects to draw A-B together. Indeed, we observed the attraction during the approaching processes of A-B (Fig. 2b, Supplementary Movie 2); meanwhile, the original displacement between A-B was gradually adjusted by a slight rotation to achieve a precise matching. On the contrary, the magnetic N-N repulsion and the capillary attraction compete between A-A (Fig. 2c). By the subtle adjustments of the relative strength of both interactions, the resultant force displayed a repulsive effect to hinder the formation of AA clusters as shown in Fig. 2d and Supplementary Movie 3. These phenomena suggested that the MSA results were determined by the strength difference between magnetic and capillary forces dependent on the

dynamically changing distance between the building blocks. Namely, the understandings of the quantified force-distance correlations are requisite to reveal the assembly mechanism.

## Calculation and measurement of capillary/magnetic forces

Therefore, we have studied the contributions of capillary and magnetic forces by calculations, simulations, and force measurements. The calculation of capillary forces was demonstrated with the example between two hydrophilic surfaces (Fig. 3a). The menisci of the side surfaces gradually merge as A-B approach into proximity; the resultant lateral force ($F_{MSA}$) is responsible for the assembly based on the force balance as shown in Eq. (1)[42]:

$$F_{MSA} = F_{capi} \cdot \cos\theta + \hat{F}_{capi} \cdot \cos\hat{\theta} = \gamma L(\cos\theta - \cos\hat{\theta}) \quad (1)$$

where $F_{capi}$ and $\hat{F}_{capi}$ are capillary forces that exert on the opposite side surfaces; $\theta$ and $\hat{\theta}$ are the angles between the capillary forces and the horizontal direction; $\gamma$ is the surface tension of the $H_2O_2$ solution; $L$ is the wetted length of the three-phase line, which is a constant value of 8 mm. During the assembly, the contour of the merged meniscus changes (marked as the red line in Fig. 3a) with the interactive distance between A-B ($x_d$), leading to the change of $\theta$ and $F_{MSA}$, accordingly; the single menisci of the opposite side surfaces (blue lines) change little. Therefore, the key to solve Eq. (1) is to build the contour functions of the merged meniscus and single meniscus to determine $\theta$ and $F_{MSA}$. We obtained the height function of a single meniscus ($Z_1$) and the merged meniscus ($Z_{total}$) in a 2D $z$-$x$ coordinate system (Supplementary Fig. 4):

$$Z_1 = Z_0 e^{-\sqrt{\frac{\rho g}{\gamma}}x} \text{(single meniscus)} \quad (2)$$

$$Z_{total} = Z_0 e^{-\sqrt{\frac{\rho g}{\gamma}}x} + \hat{Z}_0 e^{-\sqrt{\frac{\rho g}{\gamma}}(x_d-x)} \text{(merged meniscus)} \quad (3)$$

where $\rho$ is the liquid density, $g$ is the gravitational acceleration and $Z_0 = \hat{Z}_0 = 3\,mm(x = 0\,mm)$ (measured). Thus, the local heights of the menisci were calculated and simulated, which fitted well with the experimental results in Fig. 3a (i and ii). Moreover, we correlated the height functions ($Z$) with the angles ($\theta$ and $\hat{\theta}$) by the local slope of the curved surfaces:

$$\tan\theta = \frac{\partial Z_{total}}{\partial x} \quad (4)$$

which has the boundary conditions of $x_d \rightarrow \infty$, $\hat{\theta} = constant$ and $x = 0$, $\theta = \theta(x_d)$. With Eqs. (1)–(4), we calculated and plotted the numerical results of the lateral force ($F_{MSA}$) depending on the interactive distance ($x_d$), as displayed in Fig. 3b, c, which correspond well with the measured lateral forces (Supplementary Fig. 5). The dependence of lateral forces on meniscus height also matches well with previously reported experimental results[25].

The magnetic force was measured during the dynamic approaching process of A-B with a force apparatus shown in (Supplementary Fig. 6)[41]. The measurement was conducted by a cycled contact-separation process between A-B, during which the force changes exerted on A and the distance that B had been moved were in situ recorded. We have summarized the measured magnetic force, the calculated capillary forces and the resultant lateral forces between A-B and A-A in Fig. 3b, c. From the local magnification of A-A in the inset of Fig. 3c, we could observe that the capillary forces were larger than magnetic forces when the interactive distance was larger than 10.6 mm but became lower as the distance reduced, which supported the observed repulsion between A-A when the building blocks approach into proximity (Fig. 2d). Namely, the self-sorting mechanism of A-B and A-A or B-B is established by the additivity of magnetic/capillary forces

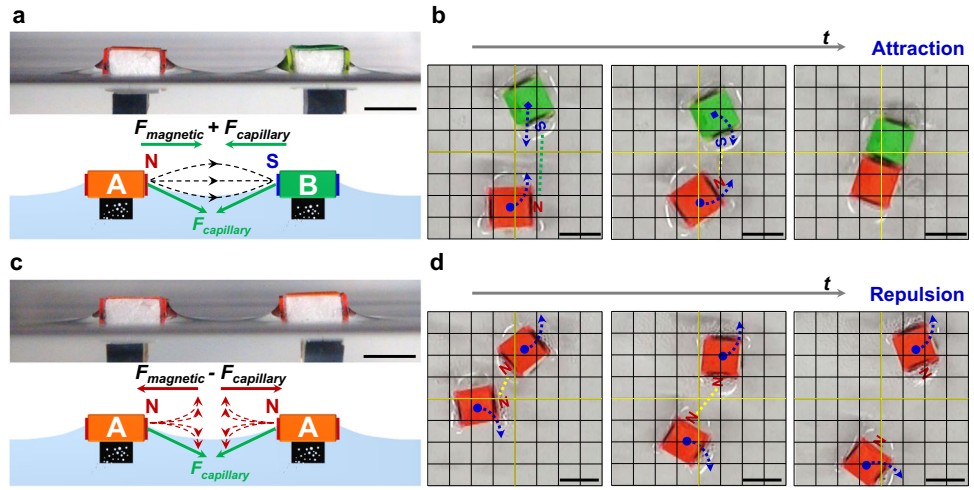

**Fig. 2 | Rough force analysis of magnetic ($F_{magnetic}$) and capillary ($F_{capillary}$) interactions underlying the self-sorting of A-B and A-A.** Photographs of **a** A-B and **c** A-A building blocks at air/water interfaces and schematic illustration of the magnetic/capillary interactions; stepwise snapshots as the building blocks of **b** A-B and **d** A-A approach. Scale bars: 8 mm.

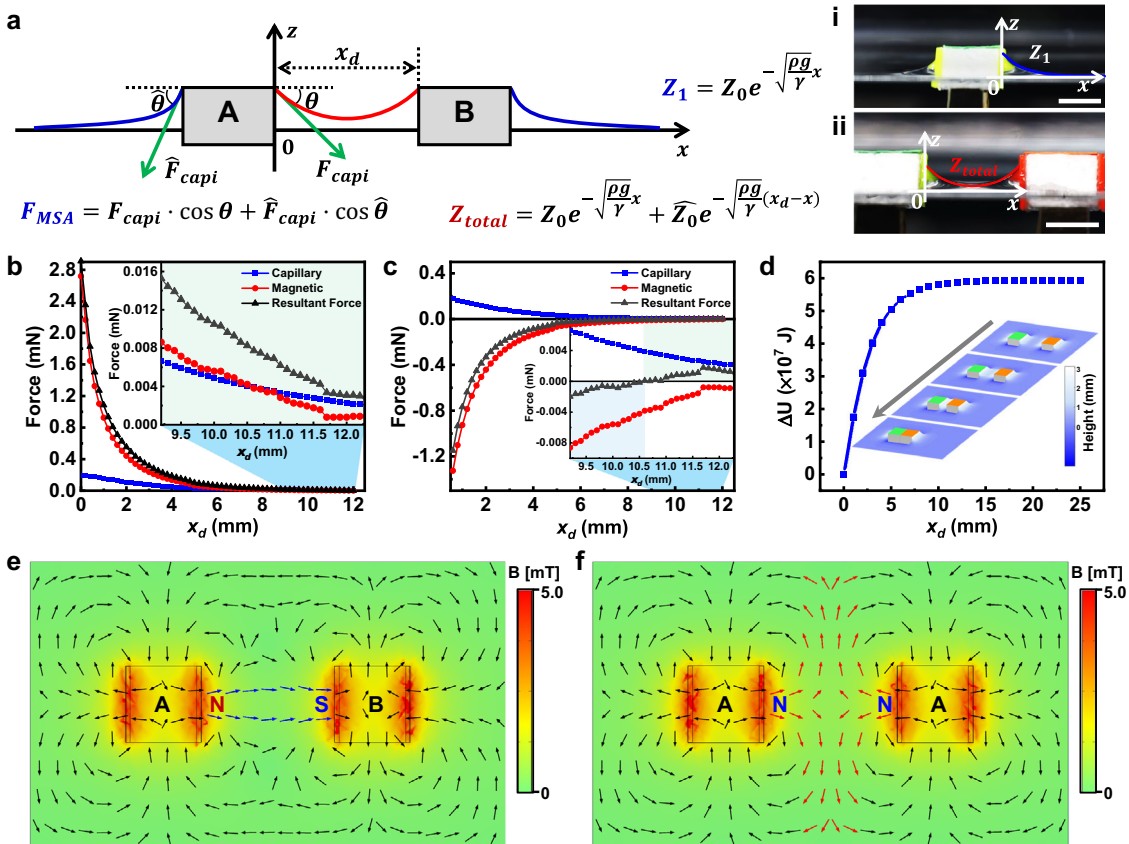

**Fig. 3 | Calculations and simulations of capillary and magnetic forces.** **a** Capillary forces induced by single ($\hat{F}_{capi}$) and merged ($F_{capi}$) menisci at the three-phase-contacting lines dependent on the angles ($\hat{\theta}$ and $\theta$) between the according menisci and the horizontal direction as the interactive distance ($x_d$) changes; the resultant lateral force for assembly is $F_{MSA}$. The height functions of $Z(x)$ to display the meniscus contours ($Z_1$: single meniscus; $Z_{total}$: merged meniscus) depending on the density ($\rho$), surface tension ($\gamma$) of the liquid, the original meniscus height ($Z_0 = \hat{Z}_0 = 3\,mm$) at $x = 0$; $g$ is the gravitational acceleration. The calculated/simulated contours are shown in the meniscus photos (i, ii). Scale bars: 5 mm. Correlations between the capillary/magnetic forces dependent on the interactive distances between **b** A-B and **c** A-A, which exhibit additive and competitive effects, respectively. **d** The calculated free energy changes ($\triangle U$) with the interactive distance ($x_d$) by integration of $F_{MSA}$. The insets show the simulated energy maps during the stepwise approaching between A-B as the interactive distance was varied ($x_d = 15, 10, 5, 0\,mm$). Simulated results of the distribution of local magnetic field between **e** A-B and **f** A-A. Source data are provided as a Source Data file.

leading to the attraction of AB, and the force difference of these two interactions resulting in the repulsion between AA or BB. The difference of the surface chemistry on A and B is thus identified via the additive effects of magnetic/capillary forces.

Furthermore, the local self-alignment was understood by the change of the free energy ($\triangle U$) dependent on the interactive distance (Fig. 3d), which was obtained by the integration of the capillary forces ($F_{MSA}\text{-}x_d$). The stepwise energy maps during the assembly of A-B were displayed as the inset of Fig. 3d by the simulation when the interactive distance was varied ($x_d = 15, 10, 5, 0\,\text{mm}$). As A-B approached and contacted, the total free energy decreased to the minimized level, which was an assumed energy condition of 'zero' potential in the simulation. Any displacements between the hydrophilic surfaces of A-B will lead to the conflicts of the adjacent hydrophilic-hydrophobic surfaces and increase the free energy. Finally, A-B underwent a self-alignment process to form an ordered dimer as confirmed by the experimental results in Fig. 2b. The distribution of the magnetic potential in cases of A-B and A-A were simulated with the methods of finite element analysis (Fig. 3e, f), which fitted the global effects of the magnetic attraction (A-B) and repulsion (A-A) well.

## Application of the self-sorting mechanism for selective assembly

To apply the above self-sorting mechanism in selective assembly, we used a model system consisting of two As and one B. Unlike the molecular self-assembly with numerous building blocks to provide averaged results, the MSA research relies on a number of parallel assembly events to provide statistical results. For example, we placed 100 containers with 100 identical groups of assembling components in an array of $10 \times 10$; the assembly was conducted under the same condition and the results in each container were summarized by displaying all the photos of the assembled structures in a similar array of 10×10, as shown in Fig. 4a, b. As expected, the addition of magnetic/capillary interactions has demonstrated the successful self-sorting of AB assembly to result in 100% ABA structures (Supplementary Movie 4, Fig. 4a). Specifically, the 100 parallel groups completed precise and selective assembly within 7 min without human intervention (Supplementary Fig. 7). As control, the conventional capillary-driven assembly without inducing the magnetic forces only led to an ABA yield of 38% (Fig. 4b, Supplementary Movie 5) and the left 62% structures were AAB, which matched with the roughly statistical estimation from the component ratio of A:B=2:1. Subsequently, the surface groups of polycations and polyanions interacted through the electrostatic attraction to chemically stabilize the AB dimer, thus allowing for lifting the structure out of the air/water interface immediately after the assembly (Fig. 4c). The reason underlying the rapid intermolecular interactions is the interfacial multivalency facilitated by the 'flexible spacing coating' beneath the polyelectrolytes according to our previous reports[18,43]. Early works of macroscopic relied on ex situ post-crosslinking methods[28,44] to stabilize the structures assembled by capillary forces, which would otherwise collapse upon leaving the air/water interface. Here, the fast interfacial adhesion by the 'flexible spacing coating' has enabled the in-situ stabilization of the MSA structures and addressed the challenge to obtain free-standing assemblies in macroscopic self-assembly[45,46]. With the above understandings, advanced chain-like structures of alternate sequences ('ABAB…'), e.g., tetramer, hexamer and octamer, were demonstrated with a good selectivity as shown in Fig. 4d and Supplementary Movie 6.

## Contribution of wettability conflicts to self-sorting

Because the magnetic forces become dominant as the building blocks approach to proximity, it remains to clarify whether only using the magnetic force is enough to realize the self-sorting. We conducted control experiments of ABA systems without inducing the wettability conflicts, namely, the four side surfaces were all hydrophilic or all hydrophobic (Fig. 5). In the all-hydrophobic cases, we observed three

'isomers' of ABA (Supplementary Movie 7): (1) the designated ABA structures only had a yield of 40% rather than the expected 100%; (2) triangle-shaped AAB and (3) line-shaped AAB, both of which had one A rotated by 90° (Fig. 5b, c). These results are attributed to the global effects of the magnetic force that exerting to all individuals[33]; in the absence of the local capillary alignment, some weak secondary magnetic forces affected the assembly results: as shown in Fig. 5a, besides the strong magnetic force ($F_{mag-N-S}$) between the two opposing N-S poles, the attraction ($F_{mag-N-2S}$) between one N pole of the top rotated A and the other two S poles of the bottom A, also exists. Meanwhile, the side surfaces are all hydrophilic or hydrophobic to result in the same menisci favorable for the attraction. In other words, the additivity of the secondary magnetic force and the capillary attraction leads to the formation of the AAB isomers (Supplementary Fig. 8). Similar results were also observed in all-hydrophilic cases (Fig. 5d, e, Supplementary Movie 8). The above control experiments have confirmed the necessity of the wettability conflicts to induce local capillary alignments: when the adjacent side surfaces of A have oppositely shaped menisci with the wettability conflicts of alternate hydrophilicity-hydrophobicity, the aggregation of AAB could be hindered because the capillary repulsion is stronger than the secondary magnetic force ($F_{mag-N-2S}$) (Supplementary Fig. 9). In addition, the mild self-propulsive motions of the building blocks driven by oxygen bubbles may provide slight disturbance to assist the local self-alignment. Taken together, both the global magnetic sorting and the local capillary alignments are requisite to demonstrate the additive effects for the self-sorting mechanism in macroscopic self-assembly.

## Discussion

We demonstrated a self-sorting strategy in macroscopic self-assembly by using the additive effects of long-ranged magnetic/capillary forces and realized selective assembly of specific structures with a yield of 100%. We integrated multiple designs to the fabrication of MSA building blocks with self-propulsive systems, wettability conflicts, and magnetic sorting for the purposes of autonomous diffusion, capillary, and magnetic-guided assembly, respectively. A self-sorting mechanism was built to identify the surface chemistry of building blocks based on the A-B attraction and the A-A repulsion resulted by the additivity and the competence between long-ranged magnetic/capillary forces, accordingly. We applied the self-sorting mechanism to realize specific assembly of ABA by excluding other isomers that were inevitable in conventional assembly with magnetic or capillary forces only. High-yield assembly of advanced structures of tetramers, hexamers and octamers have also been demonstrated. The self-sorting strategy has provided solutions to selective and precise MSA of specific structures with high yield, which could bridge the gap of self-assembly between the molecular level and the macroscopic scale to build complex supramolecular structures[47] with massive and parallel features.

The prospect of the self-sorting strategy is bright with future technological improvements on scalability and miniaturization, which is feasible owing to the high compatibility of MSA with most nano-/micro-fabrication technologies[48,49], especially on the aspect of integrating multiple materials. Ordered assemblies with high selectivity of designated components with a good assembly precision, have been in demand by diverse advanced research fields including microrobots/swarm robots[48,50], modular actuators[6,7,51], programmable encoding[52], metamaterials[53], display[54], etc., which normally require designated ordering of μm-to-mm sized components and exhibit complex, integrated, intelligent, and/or self-adaptive characteristics.

## Methods
### Fabrication of MSA building blocks
Each MSA building block consists of a cuboid expanded polystyrene (EPS) with a dimension of $8\,\text{mm} \times 7\,\text{mm} \times 5\,\text{mm}$, two magnetic plates in a hexagonal shape embedded in two thin PDMS plates

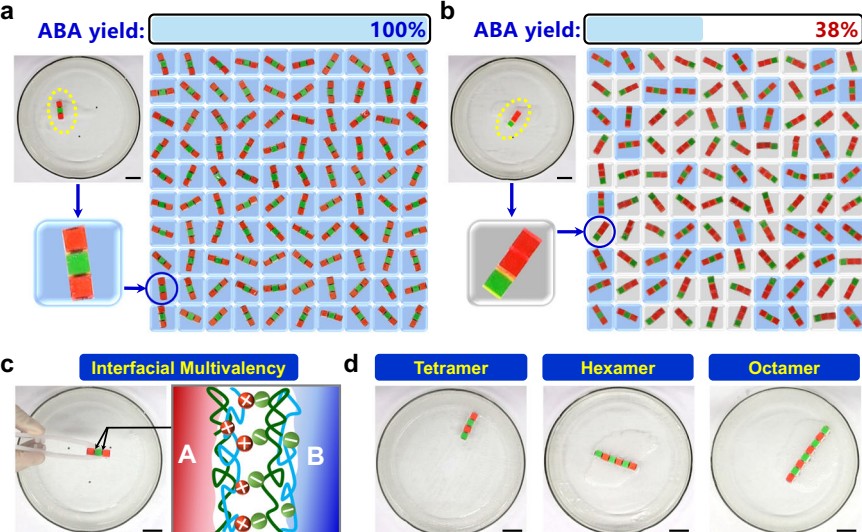

**Fig. 4 | Application of the self-sorting mechanism for selective assembly of specific structures.** Averaged MSA results of 100 parallel assembly events with two As and one B, leading to different yields of the alternate ABA structures: **a** ABA yield=100% when capillary/magnetic forces were combined; **b** ABA yield=38% when only capillary forces were present. **c** The assembled ABA structure was immediately stabilized by the interfacial multivalency of electrostatic interactions between the oppositely charged polyelectrolytes on surfaces, thus allowing for lifting ABA out of the air/water interface. **d** Feasibility of the self-sorting mechanism in fabricating advanced structures of a tetramer, a hexamer and an octamer. Scale bars: 2 cm.

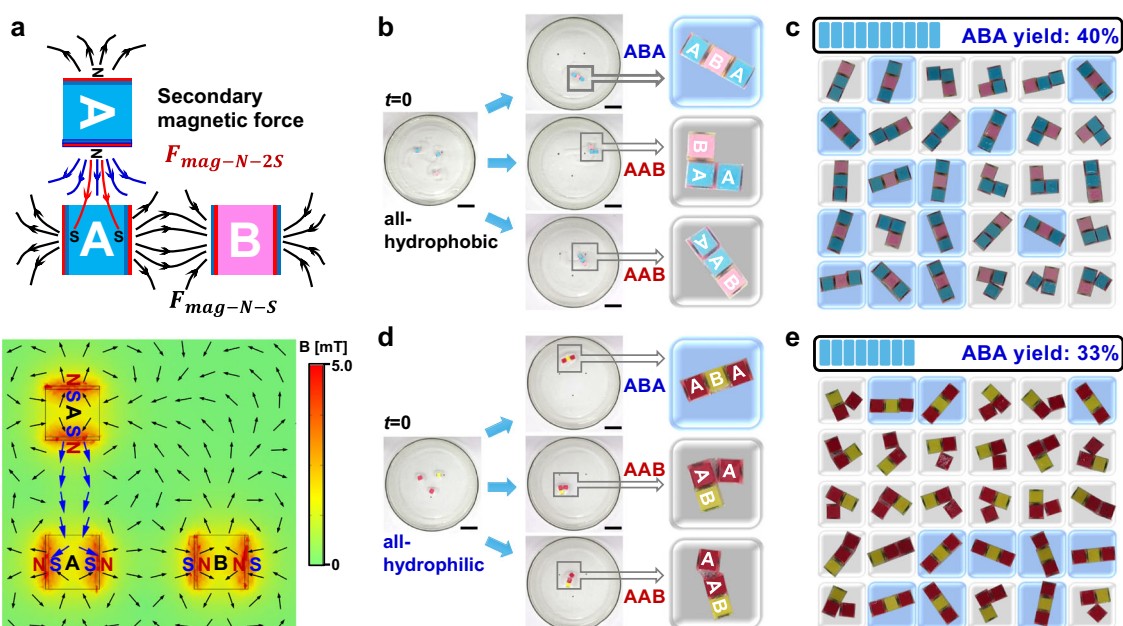

**Fig. 5 | Control experiments of MSA with isomers when lacking of local capillary self-alignments. a** Schematical illustration of AAB clusters formed by the weak secondary magnetic force ($F_{mag-N-2S}$) besides the strong opposing magnetic force of $F_{mag-N-S}$ and the bottom is the simulated distribution of local magnetic field. Photographs of three 'isomers' of ABA when all side surfaces of the building blocks are **b** hydrophobic and **d** hydrophilic, leading to the corresponding ABA yield of **c** 40% and **e** 33%, respectively. Scale bars: 3 cm.

(8 mm × 5 mm × 0.6 mm), and two glass slides (5 mm × 4 mm × 0.5 mm) deposited with rough platinum aggregates. The EPS was modified to exhibit a superhydrophobic surface with a water contact angle (WCA) of 150.9° by immersing in a dispersion of silicon dioxide nanoparticles (15 ± 5 nm) that were pre-modified with 1H, 1H, 2H, 2H-perfluorooctyltrichlorosilane. The PDMS plates embedded with magnets were modified with an LbL technique (Supplementary Section 3) to display a hydrophilic surface (WCA=28.7°). The rough platinum aggregates on glass slides were obtained by a two-step process of electrochemical deposition: the glass slides sputtered with gold

were used as the working electrode in a mixed solution of $H_2SO_4$ (aq, 0.5 mM) and $HAuCl_4$ (aq, 1 mg/mL) to be desposited for 1600 s under a constant potential of −200 mV; afterwards, the above plates were conducted with a second deposition following the same mode except changing to a mixed solution of $H_2SO_4$ (aq, 0.5 mM) and $H_2PtCl_6$ (aq, 1 mg/mL) for 2400 s (−200 mV). These components were assembled to an entity as schematically illustrated in Supplementary Fig. 1.

In all experiments with magnetic forces, building blocks that have north poles of magnets facing outward were marked as 'A' while building blocks that have south poles facing outward were marked as

'B'. In control experiments without magnetic forces, normal PDMS plates without embedding any magnets were modified similarly following the above LbL processes to obtain hydrophilic properties. In control experiments of 'all-hydrophilic' side surfaces, the commercially available EPS were modified to be hydrophilic with the above LbL processes.

## Derivation of contour functions of menisci

The menisci geometry is determined by both capillary (Young-Laplace equation) and gravitational energy; meanwhile the pressure difference results in rising of the menisci above the water surface, leading to the following equations:

$$\triangle p = \frac{2\gamma}{R} = \rho g Z \tag{5}$$

where $\triangle p$ is the pressure difference between the liquid side and air side of the menisci, $\gamma$ is the surface tension of the liquid, $R$ is the curvature radius of the menisci, $Z(x)$ is the height function of the meniscus which equals to zero as $x$ approximates to infinitely far, $g$ is the gravitational acceleration, and $\rho$ is the liquid density. Equation (5) reveals that the meniscus contour is determined by the curvature radius ($R$) when other parameters are constant for a given system. The curvature radius ($R$) can be calculated by the local curvature of the menisci ($K$) at any positions of the curved air/water surface, which equals to the reciprocal of $R$ and is dependent on several partial derivatives of the contour function ($Z$):

$$K = \frac{\left(1+\left(\frac{\partial Z}{\partial x}\right)^2\right)\frac{\partial^2 Z}{\partial y^2} - 2\frac{\partial Z}{\partial x}\cdot\frac{\partial Z}{\partial y}\cdot\frac{\partial}{\partial y}\left(\frac{\partial Z}{\partial x}\right) + \left(1+\left(\frac{\partial Z}{\partial y}\right)^2\right)\frac{\partial^2 Z}{\partial x^2}}{2\left(1+\left(\frac{\partial Z}{\partial x}\right)^2+\left(\frac{\partial Z}{\partial y}\right)^2\right)^{3/2}} = \frac{1}{R} \tag{6}$$

After correlating Eq. (5) & (6), and neglecting high-order terms[42] in Eq. (6), we obtain

$$\frac{\partial^2 Z}{\partial y^2} + \frac{\partial^2 Z}{\partial x^2} - \frac{\rho g Z}{\gamma} = 0 \tag{7}$$

Considering the assumption of infinitely long faces, namely $\frac{\partial^2 Z}{\partial y^2} = 0$, we further simplified the equation as:

$$\frac{\partial^2 Z}{\partial x^2} = \frac{\rho g Z}{\gamma} \tag{8}$$

By applying the boundary conditions of the meniscus height, i.e., $Z = 0\ mm(x \rightarrow \infty)$ and $Z_0 = 3\ mm(x = 0\ mm)$ (measured), we obtain the contour function of hydrophilic meniscus:

$$Z = Z_0 e^{-\sqrt{\frac{\rho g}{\gamma}}x} \tag{9}$$

In the case of two building blocks with a distance of $x_d$, the height function of the merged menisci was obtained by additive effects of two height functions, $Z_1$ and $Z_2$, which are shown in dashed lines to indicate virtual contours generated by A and B side surfaces. $Z_1$ fits the contour function of single meniscus in Eq. (9) while $Z_2$ can be obtained by the transformation of the function of $Z$ by translation and reflection:

$$Z_1 = \hat{Z}_0 e^{-\sqrt{\frac{\rho g}{\gamma}}(x_d - x)} \tag{10}$$

where $\hat{Z}_0 = 3\ mm(x = 0\ mm)$ (measured). Finally, we obtained the height function ($Z_{total}$) of the merged contour when the two building blocks approach into proximity by adding of the two height functions of $Z_1$ and $Z_2$:

$$Z_{total} = Z_0 e^{-\sqrt{\frac{\rho g}{\gamma}}x} + \hat{Z}_0 e^{-\sqrt{\frac{\rho g}{\gamma}}(x_d - x)} \tag{11}$$

## Measurement of magnetic forces and simulation of magnetic field

We applied a force apparatus of Dynamic Contact Angle Measuring Device and Tension Meter (DCAT21) to measure the magnetic force dependent on the distance between the magnetic plates. One magnetic plate was connected to the scale sensor at the top and the other magnetic plate was fixed on the motor at the bottom of the apparatus; their N and S poles were placed opposing each other. The measurement was conducted by a closed loop of 'approach-separation' between the two magnetic plates. The mass changes on the sensor were recorded as a mass-position curve; the peak value of the curve in the approach process indicated the contribution from the maximum magnetic attraction, which occurred when the two magnets contacted each other, namely their interactive distance ($x_d$) was zero.

## Data availability

The source data of Figs. 1b, 3b–d, Supplementary Figs. 5c, 6b, 7, and 9 are provided as a Source Data file. Source data are provided with this paper.

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

## Acknowledgements

This work was supported by the National Science Foundation for Distinguished Young Scholars (51925301) (F. S.), the National Natural Science Foundation of China (52122315, 21972008) (M. C.), Beijing Nova Program (Z201100006820021) (M. C.), the Fundamental Research Funds for the Central Universities (XK1902) (F. S.), Wanren Plan (wrjh201903) (F. S.), and Open Project of State Key Laboratory (sklssm2022) (F. S.).

## Author contributions

M.T. performed the experiments and analyzed the data. P.T. assisted the simulation and calculation. Q.Z., G.Z. and Y.L. assisted the experiments. M.C. led data analyses and wrote the manuscript. F.S. designed the research and revised the manuscript. Both M.C. and F.S. led funding acquisition, project administration and validation.

## Competing interests

The authors declare no competing interests.
