## [Peer Review File · Nature Communications]

Self-sorting in macroscopic supramolecular self-assembly via additive effects of capillary and magnetic forcesREVIEWER COMMENTS

Reviewer #1 (Remarks to the Author):

The manuscript by Tan et al has demonstrated a new self-sorting mechanism in self-assembly by coupling two long-ranged forces: magnetic, and capillary attractions. With the feasibility of engineering μm -to- mm components, they could adjust the wettability and magnetic field to realize either their additive attraction or competitive repulsion for selective assembly. They also used calculation and simulation to quantify both forces for subtle designs. Finally, the assembly system of AAB realizes a high-yield (100%) selective assembly ABA structures and other advanced structures. These results are impressive because normal wettability-induced assembly could not identify ABA from AAB based on capillary attraction. Similarly, only using magnetic attraction will cause non-specific clusters. This new self-assembly capable of self-sorting is meaningful for precise and selective bottom-up fabrication of complex structures. Therefore, I recommend this work to be published based some minor revisions.

1. The strategy of self-sorting is very clever without human intervention. I wonder whether it is possible to dissociate non-selective clusters with external energy? What about the time scale to obtain selectively assembled structures?
2. In Fig. 4c, the assembled ABA structures could be lifted directly from the air/water interface. Whitesides et al used curing polymer to form stable structures. I wonder what is the mechanism of this interfacial adhesion? Will simply modify the surfaces with interactive motifs also be possible to realize the same effects?
3. What is the height of the meniscus? Will the meniscus height influence the strength of capillary attraction?
4. The authors have measured and simulated the capillary forces in Supplementary Fig. 5. I observed good matching of both results when the components are relatively far away. What are the reasons for some deviation when the components approach closely?
5. The authors have demonstrated linear structures with this self-assembly method. I wonder if it is possible to make more complex structures?

Reviewer #2 (Remarks to the Author):

Cheng, Shi, and co-workers have studied the macroscopic supramolecular assembly (MSA) of centimeter-sized polystyrene blocks floating in aqueous solution. They used a standard approach of coating sides of the blocks with surfaces of different wettability, leading to long-range attractive capillary forces based on meniscus-matching. They set out to direct the assembly of different kinds of building blocks using different methods. Coating surfaces with positively and negatively – charged

polymers did not produce high-fidelity ordering, since the capillary forces are long-range and attractive, regardless of whether the surface is positively or negatively charged. The electrostatic repulsion between like-charged surfaces is short ranged. Thus the capillary forces dominate and selectivity is not achieved. The authors then showed that by installing magnets on the surfaces, the building blocks spontaneously assembled ABABAB chains, where A and B units have magnetic north and south poles facing outward, respectively. This success is due to the fact that both magnetic and capillary forces are similarly long-ranged. They further showed that both forces were required for this assembly, since magnetic forces alone do not have sufficient directionality to consistently select the desired interaction surfaces. These experimental observations were backed up with calculations of the energy surfaces for assembly. Overall, this seems like an interesting study and is presented clearly enough for me to be confident in the results and conclusions of the study. I do, however, have some concerns about the generality and overall impact of these findings, as follows:

1) The authors conclude that magnets are better suited for directing MSA than host-guest interactions. However, this is only really true when combined with capillary forces, due to the mismatch in length scales. As the authors have already cited, Harada et al (Nat Chem 2011) achieved high-fidelity ABABAB – type self assembly via host (cyclodextrin) / guest interactions for hydrogel particles in water. This would seem to be the more general and powerful approach, since an enormous variety of host/guest combinations exist, opening the door to more complex 3-dimensional structures. In contrast, magnets provide only a single type of recognition (N-S) and produce secondary fields that can lead to misassembly (as the authors have shown). Furthermore, they are only necessary when performing capillary-driven assembly for blocks at a liquid/water or liquid/liquid interface, which restricts patterning to two dimensions. Does capillary-driven assembly provide clear advantages over other types of assembly that would justify all of the complications and trade-offs involved in resorting to installing magnets in the building blocks?

2) How scalable is this approach? It relies on installing magnets with specific orientation on specific surfaces of each and every building block. This would seem to be hugely more labour intensive than other approaches in this field. Clearly any application would involve a very much larger number of very much smaller building blocks. What is the minimum size of building block where this could reasonably be applied? On what scale could these magnetic blocks be manufactured and at what cost? I have my doubts that this approach could ever be applied on a practical scale.

Point-to-Point Respond

To Reviewer #1 (Remarks to the Author):

The manuscript by Tan et al has demonstrated a new self-sorting mechanism in self-assembly by coupling two long-ranged forces: magnetic, and capillary attractions. With the feasibility of engineering μm -to- mm components, they could adjust the wettability and magnetic field to realize either their additive attraction or competitive repulsion for selective assembly. They also used calculation and simulation to quantify both forces for subtle designs. Finally, the assembly system of AAB realizes a high-yield (100%) selective assembly ABA structures and other advanced structures. These results are impressive because normal wettability-induced assembly could not identify ABA from AAB based on capillary attraction. Similarly, only using magnetic attraction will cause non-specific clusters. This new self-assembly capable of self-sorting is meaningful for precise and selective bottom-up fabrication of complex structures. Therefore, I recommend this work to be published based some minor revisions.

Author Reply Summary: we are thankful that the reviewer has a positive opinion about our work and provides kind suggestions. We have addressed all the following minor revisions with supports of either supplementary experiments or literatures.

Comment 1: The strategy of self-sorting is very clever without human intervention. I wonder whether it is possible to dissociate non-selective clusters with external energy? What about the time scale to obtain selectively assembled structures?

Author Reply Summary: Thanks for the suggestive comment.

Inducing appropriate energy (e.g., stirring) is possible to identify AB from AA or BB by selectively disassembling structures that have relatively weaker interactions. But the spontaneity is low because whether AA or BB has been dissociated needs to be frequently checked and corrected with human intervention.

Our self-sorting mechanism has the advantage of high spontaneity without relying on human intervention or equipment in the assembly processes: the components underwent self-propulsion, self-identification, and self-correction to realize selective and precise self-assembly. Moreover, the time of self-sorting is short to realize 100 parallel assembly events to form ABA structures shown in Fig. 4a, roughly within 7 min (Fig. R1). However, one can imagine the external-energy-involved sorting would probably take much longer

time and rely more on equipment to complete the task of 100 assembly events: one-by-one operation is time-consuming while parallel operation requires more equipment and human resources.

Fig. R1. Time-dependent assembly ratio of 100 parallel assembly events of ABA structures shown in Fig. 4a.

Specifically, the feasibility of external-energy-involved sorting as suggested by the reviewer is interpreted as follows. (1) Since AA or BB has little chemical association but mainly physically capillary attraction, external energy such as stirring or shaking could disassemble such association by breaking the menisci and lateral capillary forces that hold them. (2) Meanwhile, AB assemblies have an additional chemical connection of electrostatic interaction between oppositely charged polyelectrolytes, which provides sufficient strength to lift AB out of the air/water interface or to be stable under shaking conditions. The interfacial binding forces of AB by electrostatic interaction were measured and reported to be about one magnitude larger than that of AA or BB (Q. Zhang *et al.*, *Macromol. Rapid Comm.* **2018**, 39, 1800180). Based on the above facts, an appropriate energy such as shaking or rotation could dissociate AA or BB clusters while keeping AB intact.

As for the time scale of our self-sorting strategy, we have counted and summarized the time-dependent assembly behavior of 100 parallel ABA groups shown in Fig. 4a. As shown in Fig. R1. Over 50% groups realized selective and precise assembly in 2 min and the left ones were all assembled in totally 7 min. Especially, human intervention for either check or correction was not necessary once the assembly has started; **100 groups realized selective and precise assembly of designated structures automatically, parallelly, and rapidly in only 7 min.**

Added Text, Page 11, top:

Specifically, the 100 parallel groups completed precise and selective assembly within 7 min without human intervention (Supplementary Fig. 7).

Added Figure in Supplementary Information, Page 11, bottom:

Fig. R1 has been added as Supplementary Fig. 7.

Page 12, top:

The time to realize selective ABA assembly shown in Fig. 4a has been summarized in **Supplementary Figure 7**. Over 50% groups realized selective and precise assembly in 2 min and the left ones were all assembled in totally 7 min.

Comment 2: In Fig. 4c, the assembled ABA structures could be lifted directly from the air/water interface. Whitesides et al used curing polymer to form stable structures. I wonder what is the mechanism of this interfacial adhesion? Will simply modify the surfaces with interactive motifs also be possible to realize the same effects?

Author Reply Summary: Thanks for the suggestion of highlighting efficient interfacial molecular interactions after the self-sorting processes. Simply modifying interactive groups could not stabilize the assembled structures due to the rigidity of the PDMS surfaces. We modified a polyelectrolyte coating beneath the interactive groups to facilitate the multivalent binding for stabilization of the assembled interface according to our previous reports. The underlying mechanism is efficient multivalency between interfacial groups based on improved molecular motility by this flexible coating. Detailed reasons are interpreted as follows.

(1) Rigid μm -to- mm components fail to assemble with commonly modified surface groups.

Our previous works have demonstrated that **simply modifying interactive molecules** (host, guest molecules) like what were used in self-assembly of functionalized nanoparticles, **could hardly realize self-assembly of μm -to- mm rigid building blocks** due to limited molecular motility for efficient interfacial multivalency; a critical elastic modulus (2.5 MPa) was proposed to evaluate the assembly feasibility of certain materials (*Adv. Mater.* **2014**, 26, 3009; *Angew. Chem. Int. Ed.* **2015**, 54, 8952; *Langmuir* **2016**, 32, 3617; *Angew. Chem. Int. Ed.* **2018**, 57, 8963; *Macromol. Rapid Comm.* **2018**, 39, 1800180; *J. Mater. Chem. B* **2019**, 7, 1684; *Adv. Sci.* **2020**, 7, 2002025). Specifically, most interfacial molecular interactions between μm -to- mm building blocks are hindered by unfavorable surface properties of roughness, rigidity, low molecular motility etc. (Fig. R2): (1) normally, dynamic conditions of shaking or self-propulsion are used to propel μm -to- mm building blocks for collision and assembly, meaning occasional and short-time contact for interfacial molecular

interaction; (2) only molecules reaching the interactive distance could bind with each other and thus only minor interfacial binding events occur upon surface contacting, whose binding strength is not sufficient enough to hold large components against gravity or external energy (shaking or stirring); (3) after some minor binding events occurred, nearby molecules out of interactive distance could not overcome unfavorable factors such as high surface roughness, inhomogeneous group distribution because the deformability and flexibility of rigid surfaces are too poor to facilitate more binding events. Our modulus-dependent self-assembly results (*Angew. Chem. Int. Ed.* **2018**, *57*, 8963) demonstrated that rigid materials, e.g., polydimethylsiloxane (PDMS) with a modulus over the critical value of 2.5 MPa, could not realize self-assembly. Most reports of macroscopic self-assembly are limited to soft and self-adaptive hydrogels, which could overcome unfavorable surface effects.

Fig. R2. **a**, Schematic illustration and snapshots of directly modifying host/guest groups onto rigid PDMS surfaces leading to no assembly by shaking in water. **b**, Inducing a 'flexible spacing coating' between PDMS surfaces and interactive groups realizes interfacial assembly by identically shaking in water for 1-2 min.

(2) The strategy of 'flexible spacing coating' facilitating interfacial molecular interactions.

To address the problem of rigid building blocks failing to assemble, we have previously developed a strategy of 'flexible spacing coating', which is a sandwiched polyelectrolyte multilayer between rigid surfaces and interactive groups (M. Cheng *et al.*, *Adv. Mater.* **2014**, *26*, 3009). This coating of several micrometer, has reduced the elastic modulus from 2.5 MPa to 180 KPa, decreased the surface roughness by 67%, and exhibited a self-healing performance due to its high flowability. All these characteristics are beneficial for improving the self-adaptivity for efficient multivalency and increased binding strength. As a result, rigid building blocks underwent rapid self-assembly in 1-2 min under shaking conditions in water; the interfacial binding forces were

measured to be at least one magnitude larger than the controls without this coating. **The underlying mechanism of this strategy is efficient multivalency between interfacial groups based on improved molecular motility by a flexible coating.** Two applications have been achieved with this strategy: (1) making building blocks of rigid materials that could not assemble with normal modification of groups assemble, e.g., PDMS, plastics, glass, metal (*Adv. Sci.* **2020**, *7*, 2002025); (2) accelerating the interactive kinetics for fast stabilization of assemblies, which is especially applicable to scenes in this work with occasional and short-time surface contact.

Added Text, Page 5, bottom:

Without this coating, simply modifying interactive groups could hardly form stable assemblies in such short and dynamic contacting process.

Comment 3: What is the height of the meniscus? Will the meniscus height influence the strength of capillary attraction?

Author Reply Summary: We thank the reviewer for the suggestive comments.

Fig. R3. Schematic illustration of two kinds of meniscus height involved in our work.

There are two kinds of meniscus height involved in our work (Fig. R3). One is the **height of merged meniscus** between two assembling building blocks $Z(x_d)$, which changes with the interactive distance (x_d). The other is the meniscus height of the side surfaces not involved in assembly, i. e., the **original meniscus height** of the building blocks (Z_0): the **hydrophilic** side surfaces have a constant height of **3 mm** while the depth of concave meniscus of **hydrophobic** side surfaces is constantly **2 mm**. Generally, the capillary attraction increases with the growing of both kinds of meniscus height for the following reasons.

(1) Resultant capillary strength increases with the increase of merged meniscus height, $Z(x_d)$.

As shown in Fig. R4a, when AB approaches and x_d decreases, the horizontal composition of the capillary force, $F_{capi} \cdot \cos\theta$, increases with the reduced angle of θ ; the resultant lateral force (F_{MSA}) responsible for the

capillary-driven assembly, increases (Fig. R4b) according to our derivations and force calculations based on the meniscus height function in the manuscript. Meanwhile, we could directly observe from the snapshots of Fig. R4a that the height of the merged meniscus, $Z(x_d)$, also increases with the reducing interactive distance. Taken together, the increasing height of the merged meniscus, contributes to the increase of the capillary strength for assembly.

Fig. R4. **a**, Force analysis of building blocks in capillary-driven assembly and two snapshots displaying the height of the merged meniscus, $Z(x_d)$, with the changing interactive distance of building blocks. **b**, Calculated correlation between the lateral resultant force for assembly (F_{MSA}) and x_d .

(2) Lateral capillary strength increases with the increase of original meniscus height

In our work, the original meniscus height, Z_0 , kept constant because the material system was not changed. Theoretically, this meniscus height could be adjusted by varying the material properties such as the density, buoyancy, surface wettability etc. In our previous work (M. Xiao *et al.*, *Angew. Chem. Int. Ed.* **2015**, *54*, 8952), we have varied the density of PDMS building blocks at a water/oil interface to adjust the driving force of capillary for self-assembly. As shown in Fig. R5a-d, the meniscus height of hydrophilic side surfaces gradually increased with the increase of the PDMS density; meanwhile, the meniscus height of hydrophobic side surfaces declined. As a result, the driving force of self-assembly transformed from hydrophobic capillary attraction to hydrophilic attraction (Fig. R5a'-d'). This is because **the increased meniscus height contributed to corresponding capillary strength**.

Fig. R5. **a**, Side view photographs of PDMS building blocks at water/oil interfaces. Four side surfaces of all PDMS were modified with alternate surface wettability of hydrophilic-hydrophobic; the red dash lines represent the interfaces while the yellow solid lines display the meniscus shapes of hydrophilic side surfaces; the insets indicate meniscus shapes of hydrophobic side surfaces. **b**, Top view of assembled PDMS at water/oil interfaces: low-density PDMS underwent self-assembly via hydrophobic surfaces while high-density ones were assembled by hydrophilic capillary attraction. Adapted from *Angew. Chem. Int. Ed.* **2015**, *54*, 8952.

Added Text, Page 8, bottom:

The dependence of lateral forces on meniscus height also matches well with previously reported experimental results (Ref. 25).

Comment 4: The authors have measured and simulated the capillary forces in Supplementary Fig. 5. I observed good matching of both results when the components are relatively far away. What are the reasons for some deviation when the components approach closely?

Author Reply Summary: We are thankful about this comment. The deviation of simulated and measured/calculated capillary forces in **Supplementary Figure 5**, may be resulted by slight rotations of building blocks in the approaching process of interactive building blocks (Fig. R6). Such rotation in the x-y plane caused by the wettability conflicts of adjacent side surfaces contributed extra driving forces to alignment in the y direction for precise interfacial matching.

Fig. R6. Snapshots of top view when two MSA building blocks approached due to the capillary attraction. The dash lines marked as the edge direction at the last moment and the solid lines represented as the edge direction of the current moment. The arrow indicates slight rotations.

As shown in Fig. R6, we abstracted some snapshots from Supplementary Fig. 5 and indicated the slight rotations with arrows. In the simulation model, we mainly calculated the lateral driving forces based on one-dimensional equations in the x direction. Actually, the capillary forces from both hydrophilic and hydrophobic side surfaces increase dramatically when the two building blocks approach into proximity, which increases the driving force cooperatively and favors for long-ranged alignment to realize precise matching between the interactive surfaces. If considering such occasional and slight rotation in modeling, the boundary condition at the corner of hydrophilic/hydrophobic side surfaces becomes too complex to calculate. With the current simplification, we obtained a similar simulation/experimental trend of force-distance correlations, which is satisfactory to understand the assembly mechanism.

Added Text in Supplementary Information, Page 9, bottom:

The deviation may be caused by slight rotations (shown with arrows in Supplementary Figure 5a) contributed by wettability conflicts, which was not considered in the simulation due to model simplification and calculation.

Theoretically, the capillary forces from both hydrophilic and hydrophobic side surfaces increase dramatically when the two building blocks approach into proximity, which increases the driving force cooperatively and favors for long-ranged alignment to realize precise matching between the interactive surfaces. Besides, complex fluidic dynamics may also influence the motions.

Revised Supplementary Figure 5a and caption, Page 10, top:

Arrows indicating slight rotations are added to the revised Supplementary Figure 5a. “Blue arrows indicate slight rotations.” has been added to the caption.

Supplementary Figure 5. Measurement and calculation of capillary attraction. **a**, Snapshots of top view when two MSA building blocks approached due to the capillary attraction. Blue arrows indicate slight rotations. **b**, Photo of building blocks with the critical interactive distance (22 mm). **c**, Capillary forces obtained by the measurements of critical interactive distance (blue line) and the simulation based on contour functions (red line).

Comment 5: The authors have demonstrated linear structures with this self-assembly method. I wonder if it is possible to make more complex structures?

Author Reply Summary: Thanks for this suggestive comment. It is possible to make more complex structures by using a combination of building blocks with different wettability patterns of side surfaces. In our work, building blocks with opposite side surfaces hydrophilic-hydrophobic could assembled as a linear. If one building block has two adjacent side surfaces both with a hydrophilic wettability, this building block could act as a corner to assemble L-shaped structures.

In our previous work (*Angew. Chem. Int. Ed.* **2015**, *54*, 8952), we have designed two kinds of building blocks (Fig. R7): one kind has side surfaces with alternate wettability of hydrophilic-hydrophobic (dyed red), normally leading to linear assemblies; the other kind has adjacent side surfaces with the same wettability (dyed green), thus acting as the corner assembly site to direct the assembly of triangle or L-shaped structures. We used Marangoni flows to propel the building blocks by releasing ethanol from the cavity onto water and observed fast assembly of a triangle with one green and two red building blocks, or an L-shaped structure with one green and three red building blocks.

Fig. R7. **a**, Schematic illustration and photos of building blocks with different wettability designs of side surfaces. **b**, Snapshots of assembly of triangle and L-shaped structures.

To Reviewer #2 (Remarks to the Author):

Cheng, Shi, and co-workers have studied the macroscopic supramolecular assembly (MSA) of centimeter-sized polystyrene blocks floating in aqueous solution. They used a standard approach of coating sides of the blocks with surfaces of different wettability, leading to long-range attractive capillary forces based on meniscus-matching. They set out to direct the assembly of different kinds of building blocks using different methods. Coating surfaces with positively and negatively – charged polymers did not produce high-fidelity ordering, since the capillary forces are long-range and attractive, regardless of whether the surface is positively or negatively charged. The electrostatic repulsion between like-charged surfaces is short ranged. Thus, the capillary forces dominate and selectivity is not achieved. The authors then showed that by installing magnets on the surfaces, the building blocks spontaneously assembled ABABAB chains, where A and B units have magnetic north and south poles facing outward, respectively. This success is due to the fact that both magnetic and capillary forces are similarly long-ranged. They further showed that both forces were required for this assembly, since magnetic forces alone do not have sufficient directionality to consistently select the desired interaction surfaces. These experimental observations were backed up with calculations of the energy surfaces for assembly. Overall, this seems like an interesting study and is presented clearly enough for me to be confident in the results and conclusions of the study. I do, however, have some concerns about the generality and overall impact of these findings, as follows.

Author Reply Summary: we are thankful about the reviewer's positive and suggestive comments. We tried our best to address the questions point by point with additional experiments and literatures as follows.

Comment 1: The authors conclude that magnets are better suited for directing MSA than host-guest interactions. However, this is only really true when combined with capillary forces, due to the mismatch in length scales. As the authors have already cited, Harada et al (Nat Chem 2011) achieved high-fidelity ABABAB – type self assembly via host (cyclodextrin) / guest interactions for hydrogel particles in water. This would seem to be the more general and powerful approach, since an enormous variety of host/guest combinations exist, opening the door to more complex 3-dimensional structures. In contrast, magnets provide only a single type of recognition (N-S) and produce secondary fields that can lead to misassembly (as the authors have shown). Furthermore, they are only necessary when performing capillary-driven assembly for blocks at a liquid/water or liquid/liquid interface, which restricts patterning to two

dimensions. Does capillary-driven assembly provide clear advantages over other types of assembly that would justify all of the complications and trade-offs involved in resorting to installing magnets in the building blocks?

Author Reply Summary:

We apologize for the misunderstanding from our description to leave the impression that magnets are better suited for directing MSA than host-guest interactions. We have revised related descriptions in the text as summarized at the end of this reply.

The misunderstanding may be caused by different focus on two issues widely concerned in the study of μm -to- mm self-assembly: (1) **assembly selectivity** (e.g., selectivity in surface chemistry to form AB rather than AA or BB) and (2) **assembly precision** (mainly the interfacial matching degree between assembled surfaces). **Molecular-interaction-directed self-assembly** simply by shaking in water could realize chemical selectivity to ensure AB connections in the assembled structures while excluding AA or BB connections; however, the assembly precision of interfacial matching degree remains poor (Fig. R8), which is inevitable due to numerous possible meta-stable assemblies especially with many interactive groups and large interactive areas available on μm -to- mm components. **Solutions to address the problem of assembly precision include the capillary-driven assembly** at 2D interfaces by pre-aligning the building blocks to realize good matching between assembly surfaces, and the **self-correction strategy** by cycled assembly/disassembly processes as we previously reported (*Adv. Mater.* **2017**, *29*, 1702444).

The advantage of capillary-driven assembly in increasing the assembly precision is clearly observed in reported works (Fig. R9) over normal self-assembly methods relying on shaking or rotation; but the design and fabrication of building blocks is relatively complex as the reviewer pointed out and the chemical selectivity remains to be addressed, which is the motivation of our manuscript. **With the self-sorting mechanism, we have addressed both assembly precision and assembly selectivity.**

Nice trade-off between assembly precision, assembly selectivity, and complexity of fabrication could be achieved if using mature microfabrication technologies (e.g., template, lithography) with parallel and massive characteristics (Figs. R12 & 13). Emerging applications based on complex self-assembled structures (Fig. R14), such as microrobots, metamaterials, massive transfer of micro-LEDs, also indicates the worth to improve the precision, selectivity, and even intelligence of artificial self-assembly.

Literature/experiments in support of the above points have been provided with

detailed point-to-point explanations below.

(1) Problem of assembly precision regarding interfacial mis-matching in molecular-interaction-directed MSA.

Fig. R8. Assembly by host/guest molecular interactions through shaking in water: **a**, irregular hydrogels, and **b**, cubic polydimethylsiloxane, both of which exhibit good selectivity but obvious mis-matching between the two assembled surfaces. Self-assembly of hydrogels directed by DNA hybridization through rotation or shaking in water: **c**, hydrogels with face-specific modification of complementary strands, and **d**, hydrogels with short DNA strands. **e**, Schematic illustration of the low-precision problem in most MSA due to diverse pathways and energy-favorable meta-stable structures.

Until now, quite a few molecular interactions have been reported to direct selective assembly of μm-to-mm components, including host/guest interactions, hydrogen bonding, DNA hybridization, electrostatic interactions, metal-ligand coordination etc., as shown in Fig. R8a-d with some examples. Generally, **rotation or shaking of building blocks immersed in water** is a common

method to trigger the collision and assembly, which however has the **long-existing problem of low precision** with poor interfacial matching (Fig. R8e). The reasons are: (1) self-assembly is “relatively insensitive to errors in registration components” (*Self-assembly at many scales*. in Page 255, Chapter 6 of *Bionanotechnology—lessons from nature*. by D. Godsell, Wiley-Liss, Hoboken, NJ, **2004**); especially when the component size increases, more assembly pathways and energy-favorable meta-stable assemblies appear; (2) rough shaking or rotation processes in normal 3D fluidic conditions could hardly realize directional assembly, resulting in poor geometry ordering.

(2) A solution of capillary pre-alignment to improve interfacial matching.

Solutions to improve the above interfacial mis-matching for good ordering mainly include **capillary alignment** by confining the assembly in two-dimensional interfaces and applying long-ranged capillary forces to align macroscopic components, and self-correction by cycled assembly/disassembly processes of the interface (Fig. R9a). With capillary-driven assembly, we have previously demonstrated several precise assembly examples of millimeter-scaled building blocks at interfaces of water/oil or air/water by combining capillary alignment and molecular recognition (Fig. R9b,c). However, the selectivity of surface chemistry remains unsolved as we analyzed in the introduction of this work, which leads to the motivation of this self-sorting mechanism.

Fig. R9. **a**, Strategies to improve interfacial mis-matching in MSA. **b**, Schematic illustration of capillary alignment for precise assembly and **c**, an example of millimeter-scaled PDMS components.

(3) Control experiments of this work to clarify the assembly precision issue.

The design of our current work includes two confinements to improve the assembly precision: (1) wettability conflicts to induce capillary alignment; (2) reducing assembly possibilities by limiting the assembly from 3D space to a 2D plane. **To clarify whether these two designs are necessary for precise assembly, we have conducted two control experiments as follows.**

Fig. R10. **a**, Schematic illustration of chemical composites and fabrication procedure of hydrophilic building blocks with either positive or negative charges. **b**, Snapshots of 2D assembly results at the air/water interface driven by stirring and the formed structure has the interfacial mis-matching problem.

Control 1: 2D assembly without wettability conflicts. Specifically, we have prepared building blocks (density: 0.81 g/cm^3) without wettability conflicts based on chemically isotropic poly(hydroxyethyl methacrylate) (PHEMA) hydrogels, as shown in Fig. R10a. The hydrogels were prepared by a template method through thermally triggered copolymerization; dyes were added to distinguish the surface chemistry induced by charged monomers: positively (red) or negatively (blue) charged PHEMA hydrogels. Then they were adhered onto EPS for 2D assembly at air/water interfaces. The complementarily charged hydrogels underwent rapid assembly upon collision and contact under stirring conditions. **Even though good chemical selectivity was observed with alternate red-blue connections, mis-matching between two assembled surfaces exists** (Fig. R10b). This is because the adjacent side surfaces have

no wettability conflicts and such mis-matched states are also energy-favorable. On the contrary, the wettability conflicts in our manuscript created totally opposite menisci between adjacent side surfaces, which will result in high-energy twisted surfaces if slightly mis-matched. **Therefore, the design of wettability conflicts is requisite for precise assembly.**

Fig. R11. **a**, Schematic illustration of the chemical composites and fabrication procedure of isotropic hydrogel building blocks with either positive or negative charges. **b**, Snapshots of 3D assembly in water after shaking process, resulting in selectively connected components but poor matching degree.

Control 2: 3D assembly of isotropic components in water. Following a similar copolymerization and template method, we prepared chemically isotropic cubic hydrogels as building blocks, which have either positive (red) or negative (blue) surface charges and a larger density than that of water (Fig. R11a). Instead of floating on water surfaces, these hydrogels were always immersed in water at the bottom of the container. Then they underwent self-assembly following a common shanking process just like most reports of molecular-interaction-driven self-assembly. After shaking for 1 min, the hydrogels assembled into a cluster with good chemical selectivity; **however, almost every assembled interface displays low assembly precision with mis-matching** (Fig. R11b), which is quite common in most reported MSA works. The underlying reason is that molecular interactions (electrostatic attraction between polyelectrolytes, host/guest recognition etc.) are too short-ranged to align μm -to- mm building blocks for precise matching. Therefore, **long-ranged forces (e.g., capillary,**

magnetic forces) in our manuscript is a solution for high-precision assembly by pre-alignment. Molecular interactions are also useful to stabilize the assembled structures after alignment by long-ranged forces; otherwise, the assemblies would collapse if leaving the 2D interfaces. Considering that the assemblies are mainly 2D structures, **one potential strategy to fabricate 3D ordered structures is stacking such chemically stabilized 2D structures layer by layer.**

Added Text, Page 2, at top, middle, and bottom of the page:

(top) As the assembly scales up with many pathways and meta-stable structures, the challenge to selectively obtain designated structures is increasing together with the problem of low assembly precision¹⁴.

(middle) The assembly precision of the above capillary-driven MSA is satisfactory with assembled surfaces well-matched; however, these MSA methods only demonstrated the selectivity in the surface wettability.

(bottom) Although molecular interactions (e.g., host/guest recognition) are chemically specific^{30, 31}, these molecular-level interactions are too short-ranged to align μm -to- mm objects and the assembly precision remains poor with interfacial mis-matching^{2, 8, 17, 18}.

Page 6, middle of the page:

However, the selectivity of the electrostatic attraction between A-B and the electrostatic repulsion between A-A, can not distinguish these assemblies of AB or AA at macroscopic scales because the molecular-leveled electrostatic interactions are much weaker than the long-ranged capillary forces.

Comment 2: How scalable is this approach? It relies on installing magnets with specific orientation on specific surfaces of each and every building block. This would seem to be hugely more labour intensive than other approaches in this field. Clearly any application would involve a very much larger number of very much smaller building blocks. What is the minimum size of building block where this could reasonably be applied? On what scale could these magnetic blocks be manufactured and at what cost? I have my doubts that this approach could ever be applied on a practical scale.

Author Reply Summary:

We are very thankful about these suggestive and inspirational comments from the reviewer. We totally agree with the reviewer that the scalability, size limit, cost, and practical applications of this newly developed approach are important for its further development. Our considerations have been summarized here

followed by point-to-point explanation.

Scalability & size: In this work, we use millimeter-sized building blocks to demonstrate the design principle of selective and precise artificial self-assembly; **the current cost is roughly estimated to be US\$0.423 per building block**, which could be further reduced by using mature scalable template or lithography methods. Indeed, our approach is compatible with current technologies which have scalable, precise, and miniaturized features, e.g., template molding, lithography, heterogeneous additive manufacture/3D printing, two photon techniques. Taking a specific lithographic example, **the reported maximum scalability** of a system integrated with multiple materials analogous to our building blocks is **over one million per four-inch wafer with the minimum size on the order of 100 μm** at an averaged cost of **US\$0.001 per robot** (M. Z. Miskin *et al.*, *Nature* **2020**, *584*, 557).

Application & prospect: Our approach matches well with the trend of self-assembly systems towards emerging advanced applications such as **microrobot/swarm robots, soft modular actuator, encoding, metamaterials, massive transfer for micro-LED display**, which normally require designated ordering of μm -to- mm sized components and exhibit complex, integrated, intelligent, and/or self-adaptative characteristics.

The followings are point-to-point explanations to the above summary:

(1) Feasibility to scale up our approach by combining with micro-/nano-fabrication.

Compared with molecular assembly, our building blocks have a unique feature of stepwise integrating multiple materials, which is analogous to micro-/nano-fabrication in semiconductor industry or micro-robotics. The scalability of this approach could be realized by **template methods** as we demonstrated in Supplementary Fig. 3 for **millimeter-sized components**, and/or by **multi-steps lithography methods** for **micrometer-sized components**, which has a very nice example reported by M. Z. Miskin *et al.* recently (*Nature* **2020**, *584*, 557). Their micrometer-sized robot has more complex structures consisting of multiple components and materials (Fig. R12a) including circuits and actuators. As shown in Fig. R12b,c, they applied a **17-step procedure** with standard doping, lithography, and metallization to create the robot's onboard circuitry; the resulted robot bodies have a maximum thickness of 5 μm , and width and length dimensions of 40 μm and 40 μm or 40 μm and 70 μm , respectively. The output

is about **one million robots on one four-inch wafer** (Fig. R12d), which shows a scalable feature owing to parallel manufacturing features of lithography methods at microscale.

Fig. R12. **a**, Optical image of a microscopic robot. It has two parts: a body with internal electronics and legs that actuate. For the work here, the electronics are simple circuits made from silicon p–n junctions and metal interconnects, encapsulated between a layer of silicon dioxide and a layer of SU-8 photoresist. The legs are made from a new class of voltage-controlled actuators called SEAs and rigid SU-8 panels. The panels control the folded shape of the leg while the SEAs produce motion. **b**, By directing laser light to photovoltaics that alternately bias the front and back legs, the robot walks along patterned surfaces. **c**, A real robot walking across a surface. **d**, Optical image of a chip with thousands of robots on it. The chip was cut from a four-inch wafer with approximately one million microscopic robots on its surface. All images are adapted from *Nature* **2020**, 584, 557.

Inspired by this work, we have designed a fabrication procedure for scalable manufacture of building blocks as schematically illustrated in Fig. R13. Hydrophobic photoresist (SU8) is used as the main body of the building blocks. Positive photoresist acts as protecting layers for selective metal deposition or surface modification to induce hydrophilicity. Specifically, anisotropic metal deposition of designated surfaces could be realized by subtle angle adjustments according to reports from O. Velev’s group (Han *et al.*, *Sci. Adv.* **2017**, 3, e1701108; C. Shields *et al.*, *Soft Matter* **2013**, 9, 9219). Steps (1)-(4) is used for the coating of a magnetic cobalt layer on top together with surface modification; Steps (5)-(7) could induce platinum layers for propulsion. To coat

another magnetic layer on the opposite side surface, the pick & place method widely applied in semiconductor industry (J. Rogers *et al.*, *Nat. Mater.* **2006**, *5*, 33) is proposed to expose the bottom surface adhered on the scarifying layer for further deposition of a cobalt layer following Steps (8)-(12). Thus, **scalable fabrication of building blocks for selective MSA is possible by using lithography methods.**

Fig. R13. Schematic illustration of scalable fabrication of micrometer-sized building blocks applicable for selective MSA.

(2) Issues of size limit, cost, and practical application of this approach.

As the size of the building blocks reduces, the limitation lies in the magnet and its magnetic performance. The smallest magnetically actuated propeller has a filament diameter of approximately 70 nm reported by P. Fischer's group (D. Schamel *et al.*, *ACS Nano* **2014**, *8*, 8794). By using a technology of micellar nanolithography, they obtained helical nickel propellers of 400-nm long with a pitch of 100 nm. These propellers could be magnetized (magnetization: 1.13×10^{-6} emu/mm²) and propelled under strong magnetic fields (about 100 Oe) against Brownian motions in viscous fluids. **Their results imply the feasibility to fabricate nanosized building blocks with heterogenous materials based on subtle nanofabrication methods.**

As for the **manufacture cost** of microscale components with complex structures, M. Z. Miskin *et al.* assumed a 180-nm CMOS lithography performed at a foundry for estimation (*Nature* **2020**, *584*, 557): with a baseline number of US\$10 per cm² and a density of 10,000 robots per cm², microrobots with roughly 100 µm on a side, with a clock, sensors and a programmable controller,

would cost approximately US\$0.001.

For **practical applications**, integration of designated μm -to- mm components is meaningful for quite a few advanced applications with some examples as follows (Fig. R14):

- microrobot/swarm robots (M. Z. Miskin *et al.* *Nature* **2020**, 584, 557; M. Rubenstein *et al.*, *Science* **2014**, 345, 795)
- soft modular actuator (D. Aukes *et al.*, *Adv. Mater.* **2021**, 2005906)
- programmable dynamic assembly or encoding (M. Sittler *et al.*, *Sci. Adv.* **2017**, 3, e1602522; J. Sessler *et al.*, *Adv. Mater.* **2018**, 30, 1705480)
- metamaterials (M. Hecke *et al.*, *Nature* **2016**, 535, 529)
- massive transfer for micro-LED display (G. Whitesides *et al.*, *Science* **2002**, 296, 323; H. Jacob *et al.*, *Adv. Mater.* **2015**, 27, 3661; eLux, Inc., <https://www.eluxdisplay.com>)

a ➤ **microrobot/swarm robots**

M. Z. Miskin *et al.*, *Nature* **2020**, 584, 557.

M. Rubenstein *et al.*, *Science* **2014**, 345, 795.

b ➤ **Soft modular actuator**

D. Aukes *et al.*, *Adv. Mater.* **2021**, 2005906.

c ➤ **Programmable encoding**

J. Sessler *et al.*, *Adv. Mater.* **2018**, 30, 1705480.

d ➤ **Massive transfer for display**

G. Whitesides *et al.*, *Science* **2002**, 296, 323.

e ➤ **Metamaterials**

M. Hecke *et al.*, *Nature* **2016**, 535, 529.

Fig. R14. Application examples of assembled μm -to- mm building blocks for **a**, study of swarm behaviors, **b**, design of modular actuators consisting of rigid/soft segments, **c**, programmable encoding of modular pixels, **d**, massive transfer of LEDs for display, and **e**, metamaterials.

Subtle strategies to identify specific individuals and then arrange them on demand, remain lacking. The self-sorting mechanism we demonstrate in this work may provide a solution to these research fields.

Added Text, Page 14, top:

The prospect of the self-sorting strategy is bright with future technological improvements on scalability and miniaturization, which is feasible owing to the high compatibility of MSA with most nano-/micro-fabrication technologies⁴⁸⁻⁵⁰, especially on the aspect of integrating multiple materials. Ordered assemblies with high selectivity of designated components with a good assembly precision, have been in demand by diverse advanced research fields including microrobots/swarm robots^{48, 51}, modular actuators^{6, 7, 52}, programmable encoding⁵³, metamaterials⁵⁴, display⁵⁵ etc., which normally require designated ordering of μm -to- mm sized components and exhibit complex, integrated, intelligent, and/or self-adaptative characteristics.

REVIEWERS' COMMENTS

Reviewer #2 (Remarks to the Author):

The authors have nicely addressed my concerns. The advantages of meniscus-based approaches in assembly precision are clear to me now, and the avenues for scaling up seem reasonable. It's a shame that manuscript does not contain more of the highly informative discussion in the rebuttal letter, although I suppose there are always space limitations.

My final comment might be to reconsider use of the word "cooperative". At least in biophysics, cooperative is used when energies are NOT additive. For example, if two sites bind ligands cooperatively, then binding at one site changes the energy of binding at the second site. In this case, capillary forces do not alter magnetic forces, and vice versa. I might suggest replacing "cooperative effects" with "additive effects", as the two energies are, in fact, additive.

Reviewer #4 (Remarks to the Author):

This elegant paper presents a systematic study on the supramolecular self-assembly of binary macroscopic cubes with a high-level of self-sorting control. The clever use of the cooperative effect of capillary and magnetic forces leads to the formation of trimer, tetramer, hexamer and octamer with defined sequences at high yield. The description of the work is clear, and the findings are well supported by experimental observations, calculations and simulations. Overall, this is an important work that is deserved for publications in Nature Communications. Thus, I support the acceptance of this manuscript enthusiastically.

One minor comment: for statistical analysis, the authors conduct experiments using 100 containers with 100 identical groups of assembling components. The system, however, does not involve a large number of the assembling building blocks. It is my curiosity about the effect of cube concentrations on the self-sorting assembly process. Will a high concentration of the assembling components reduce the yield of the trimer, tetramer, etc?

Response to Reviewer's Comments

We thank all Reviewers for their kind suggestive comments. The manuscript has been carefully revised accordingly. The point-to-point reply has been summarized as follows.

Response to Reviewer #2: Page 2-Page 4;

Response to Reviewer #4: Page 5-Page 6.

REVIEWERS' COMMENTS & POINT-TO-POINT REPLY

Reviewer #2 (Remarks to the Author):

The authors have nicely addressed my concerns. The advantages of meniscus-based approaches in assembly precision are clear to me now, and the avenues for scaling up seem reasonable. It's a shame that manuscript does not contain more of the highly informative discussion in the rebuttal letter, although I suppose there are always space limitations.

My final comment might be to reconsider use of the word "cooperative". At least in biophysics, cooperative is used when energies are NOT additive. For example, if two sites bind ligands cooperatively, then binding at one site changes the energy of binding at the second site. In this case, capillary forces do not alter magnetic forces, and vice versa. I might suggest replacing "cooperative effects" with "additive effects", as the two energies are, in fact, additive.

Author Reply:

Thank you for the kind suggestion. Indeed, we agree with the reviewer that the term "cooperative effects" has been normally referred to interplay of coupled interactions beyond simple additivity in both biological systems and supramolecular systems (J. Stoddart *et al.*, *Acc. Chem. Res.* **2005**, *38*, 723; H. Anderson *et al.*, *Angew. Chem. Int. Ed.* **2009**, *48*, 7488). To avoid confusion, we have revised "cooperative effects" with "additive effects" as suggested. Besides, descriptions or discussions related with cooperativity have also been revised both in the text and in the Supplementary Information. The detailed revisions in the text are listed as follows, which have also been highlighted with a yellow background in the marked version of manuscript.

- (Before revise) Self-sorting in macroscopic supramolecular self-assembly via cooperative effects of capillary and magnetic forces.

(After revise) Self-sorting in macroscopic supramolecular self-assembly via additive effects of capillary and magnetic forces.
- (Before revise) As a result, A-A or B-B repelled each other upon approaching due to the negative cooperation of the stronger N-N or S-S repulsion than the hydrophilic attraction.

(After revise) As a result, A-A or B-B repelled each other upon approaching due to the competition of the stronger N-N or S-S repulsion than the hydrophilic attraction.
- (Before revise) the lateral forces between A-B include the N-S magnetic attraction

and the capillary attraction, which **cooperate positively** to draw A-B together.

(After revise) the lateral forces between A-B include the N-S magnetic attraction and the capillary attraction, which **generate additive effects** to draw A-B together.

- (Before revise) The difference of the surface chemistry on A and B is thus identified via the **cooperative** effects of magnetic/capillary forces.

(After revise) The difference of the surface chemistry on A and B is thus identified via the **additive** effects of magnetic/capillary forces.
- (Before revise) **b**, A-B and **c**, A-A, which exhibit **positively and negatively cooperative** effects, respectively.

(After revise) **b**, A-B and **c**, A-A, which exhibit which exhibit **additive and competitive** effects, respectively.
- (Before revise) 100% when capillary/magnetic forces were **cooperated**.

(After revise) 100% when capillary/magnetic forces were **combined**.
- (Before revise) the **cooperation** of magnetic/capillary interactions has demonstrated the successful self-sorting of AB assembly to result in 100% ABA structures.

(After revise) the **addition** of magnetic/capillary interactions has demonstrated the successful self-sorting of AB assembly to result in 100% ABA structures.
- (Before revise) both the global magnetic sorting and the local capillary alignments are requisite to demonstrate the **cooperative** effects for the self-sorting mechanism in macroscopic self-assembly.

(After revise) both the global magnetic sorting and the local capillary alignments are requisite to demonstrate the **additive** effects for the self-sorting mechanism in macroscopic self-assembly.
- (Before revise) We demonstrated a self-sorting strategy in macroscopic self-assembly by using the **cooperative** effects of long-ranged magnetic/capillary forces and realized selective assembly of specific structures with a yield of 100%.

(After revise) We demonstrated a self-sorting strategy in macroscopic self-assembly by using the **additive** effects of long-ranged magnetic/capillary forces and realized selective assembly of specific structures with a yield of 100%.

Reviewer #4 (Remarks to the Author):

This elegant paper presents a systematic study on the supramolecular self-assembly of binary macroscopic cubes with a high-level of self-sorting control. The clever use of the cooperative effect of capillary and magnetic forces leads to the formation of trimer, tetramer, hexamer and octamer with defined sequences at high yield. The description of the work is clear, and the findings are well supported by experimental observations, calculations and simulations. Overall, this is an important work that is deserved for publications in Nature Communications. Thus, I support the acceptance of this manuscript enthusiastically.

One minor comment: for statistical analysis, the authors conduct experiments using 100 containers with 100 identical groups of assembling components. The system, however, does not involve a large number of the assembling building blocks. It is my curiosity about the effect of cube concentrations on the self-sorting assembly process. Will a high concentration of the assembling components reduce the yield of the trimer, tetramer, etc?

Author Reply:

Fig. R1. **a**, Symmetric design of building components with alternate hydrophilic-hydrophobic surface wettability of side surfaces and magnetic plates attached on two opposite hydrophilic surfaces. **b**, One dimensional chain growth mechanism along the hydrophilic side surfaces as the active binding sites. Experimental results of linear structures with different length formed by **c**, 3 components (A:B ratio=2:1), and **d-f**, 4-8 components (1:1).

We appreciate the reviewer for positive opinions about our work and suggestive comments. With the current symmetric design of building components (**Fig. R1a**), we always obtain a linear structure with sufficient assembly time at an appropriate component ratio. This is because each cuboid component has two opposite side surfaces as the 'binding sites' to connect with other components, thus leading to a chain-like growth pattern as the component number increases (**Fig. R1b**). When we increased the

component number from 3 to 8 with a close ratio to 1:1, they always assembled into a linear structure in the end (**Fig. R1c-f**) by adding new components at two end positions. Even though, the chemical selectivity is maintained with strictly alternate ABAB... connections instead of AA or BB owing to the self-sorting mechanism. The above 'chain growth' phenomenon in macroscopic self-assembly is similar to the process of 'chain propagation' mechanism in free radical polymerization that the polymer chain keeps growing until a termination mechanism functions. Another similar process is the DNA transcription with a strict sequence growing along the chain, which also stops when a transcription terminator appears. By learning from the mechanism of polymerization or DNA transcription, **the key to obtain more designated assemblies rather than longer chains when increasing the component concentration, is a termination mechanism.**

Fig. R2. **a**, Design of 'Hydrophobic Terminator' to stop chain growth of ABA trimer. **b**, Building component with anisotropic wettability (green) to direct assembly of isotropic components (red). **c**, Assembly results of triangle or L-shaped structures.

To realize high-yield trimers with increased component concentration, we propose a 'Hydrophobic Terminator' of Component A with an asymmetric wettability design. As schematically shown in **Fig. R2a**, three side surfaces of Component A are hydrophobic and only one side surface is hydrophilic available for assembly; Component B is still symmetric with two hydrophilic side surfaces. When placing such combinations of A and B (ratio=2:1) at a high concentration, high-yield ABA trimers could be obtained rather than long chains for the following reasons. (1) **The 'binding sites' of assembled Component**

A could hardly connect with another B. Even though the magnetic attraction between AB still exists, the capillary repulsion caused by opposite menisci from hydrophobic (A) and hydrophilic (B) surfaces could be advantageous over by lowering the magnetic strength. Thus, both ends of the formed ABA trimer are 'terminated' to prevent chain growth. The left components could be assembled into more ABA trimers, contributing to a high yield. (2) **Assembly directed by hydrophobic surfaces could be avoided.** The component density is so low that the menisci of hydrophobic surfaces are weak, which means weak hydrophobic capillary attraction. Meanwhile, magnetic plates are attached on hydrophilic surfaces to favor for hydrophilic attraction in priority; self-propulsive forces from the continuously releasing bubbles can act as disturbance to remove weak aggregates.

The above design based on asymmetric wettability design is feasible with a demonstration experiment in **Fig. R2c**. The red components have symmetric wettability with two opposite side surfaces hydrophilic and the other two hydrophobic, meaning they could only assemble into linear chain structures following the 1D growth direction. When we slightly changed the green components by modifying two adjacent side surfaces as hydrophilic and the other two hydrophobic, the assembly direction could be changed. The green component acts as a growth site to induce the assembly of red components along the corner, leading to either triangle or L-shaped structures. Hence, tailoring the wettability of components could tailor the assembly geometry, indicating the feasibility of the design of 'Hydrophobic Terminator' in obtaining high-yield ABA trimers when increasing the component concentration.